# Sharing images of children on social media: British motherhood influencers and the privacy paradox

Katherine Baxter[ID][1], Barbara Czarnecka[2]*

**1** Liverpool Hope Business School, Liverpool Hope University, Liverpool, United Kingdom, **2** London South Bank University Business School, London South Bank University, London, United Kingdom

* czarnecb@lsbu.ac.uk

**Data Availability Statement:** Research data for Study 1 and Study 2 are available from LSBU Public Repository via the following link: https://doi.org/10.18744/lsbu.95xy3.

## Abstract

This study examines the extent to which popular British motherhood influencers infringe on their children's privacy by posting images of them online. We conducted a content analysis of 5,253 Instagram posts from ten UK-based influencers, supplemented by self-reported data from these influencers. This represents the first comprehensive analysis of actual sharing practices in the British motherhood influencer industry, linking observed behavior with self-reported perceptions. Children appeared in over 75% of the posts (3,917), though the proportion of posts containing embarrassing, intimate, or revealing content was relatively low (11.5%). Notably, sponsorships and product advertisements were present in 46.4% of posts featuring children, indicating that children's images are frequently used for financial gain. Despite this, post popularity did not vary based on the inclusion of children, as posts featuring children did not receive more likes than those without. Influencers reported strong trust in online safety on Instagram, and reported indifference or willingness to sharenting suggesting that sharing images of their children may be a deliberate strategy rather than an accidental act. Half of the influencers did not accurately estimate their past sharenting behavior. This study extends the existing body of knowledge on sharenting behaviour and the privacy paradox by establishing a foundation of parents' real-world posting habits and connecting them to their beliefs about publicly sharing their children's images in the UK context. The findings do not strongly support the privacy paradox in this sample.

## Introduction

Over the past two decades, social media influencers have become a popular communication strategy on various social networking sites worldwide [1,2] raising concerns about privacy, particularly in the context of sharing images of children. The material posted by these social media influencers is frequently shared for financial benefits, including advertising and sponsored content [3–6]. Among the different types of social media influencers, motherhood influencers have become one of the more prolific categories of influencers [7–9]. Motherhood influencers have been defined as: *'a mommy (mum/mother) blogger who uses social networking*

**Funding:** The author(s) received no specific funding for this work.

**Competing interests:** The authors have declared that no competing interests exist.

*sites as her blogging platform'* [10] and this type of sharing has been labelled 'sharenting' [7]. Motherhood influencer share images featuring their children and families endorsing brands and products or portraying important key themes experienced in parenthood. However, motherhood influencers have been heavily criticised for sharing too much private and intimate content, especially videos and images of their children and often for commercial gain, prompting debates about ethical and privacy implications [11–13]. As children cannot consent to the sharing images and life events online, there is a growing concern that such practices are unethical, inappropriate, and violate privacy boundaries [14–17]. Some forms of 'sharenting' intrude on children's privacy and may have potential future psychological impacts, particularly as the child gains a large, and possibly unwanted, social media following [18]. Other issues that have occurred with 'sharenting' include children's images being misused, potential information getting into the hands of paedophiles, digital kidnapping, untracked monitoring of children for commercial use by organisations, and the unlawful use of children's images to sell products [19]. It may also contribute to future bullying or harassment for the child and is difficult to remove due to the eternal nature of the social media footprint and digital permanency [20–22]. It has even been suggested that for motherhood influencers, commercial opportunities can often supersede any thoughts or concerns for privacy [23–25]. In France, this issue has been taken so seriously that there has been a recent law passed that gives children who are included in influencers' posts 'the right to be forgotten' whereby at the child's request any content can be removed entirely from the social media platform [26].

Despite these significant concerns about children's privacy [10,27,28], there is no systematic evidence so far to demonstrate the extent to which motherhood influencers share content featuring their children and how the sharing behaviour relates to privacy concerns. Many studies of social media influencer behaviour make claims that "children play a significant role within their influencer mothers' content as they are continually being depicted in photos and videos on the momfluencers' profiles" [29, p. 304] but the extent to which children are featured in social media content has not been systematically examined.

Most studies on motherhood influencers and children's privacy published up to this point have been based on self-reported measures of perceived sharing behaviour rather than measures of actual behaviour (see Table 1 for a detailed review of recent literature) [30]. Such studies often make claims for the existence of a 'privacy paradox' which is defined as a dichotomy in how a person intends to protect their online privacy versus how they behave online [31,32]. Although valuable, these studies allow only for partial interpretations of sharenting behaviour due to limitations of self-reported measures [33]. As Solove [31] argues, the privacy paradox that so many studies assert, in fact does not exist, because 'privacy paradox' claims are usually due to the methodology of measuring the paradox. Indeed, Dienlin, Masur and Trepte [34] found evidence against the privacy paradox in a longitudinal study based on a within-subject study design with individual consumers in Germany. Similarly, Meier and Kramer [35] also found no support for privacy paradox.

After reviewing past studies on motherhood influencers and privacy paradox, we identified an important gap that our study addresses, namely lack of studies that examined behaviour rather than perceptions of behaviour. Most studies examining sharenting behaviour measured perceptions rather than behaviour and therefore, we follow the suggestion of Solove [31] and apply a different methodological approach, namely a two-method approach to study the privacy paradox of motherhood influencers. Specifically, we first apply content analysis to examine sharenting behaviour and then compare the content to reported privacy perceptions collected via surveys. By applying such mixed-method approach, this study addresses the following aims:: first to examine the extent to which selected British motherhood influencers share social media content featuring their children, and second, to explore how the actual

**Table 1. Summary of recent studies on sharenting.**

| Study | Sample size | Country | Research question | Method |
|---|---|---|---|---|
| Beuckels, Hudders, Vanwesenbeeck, & Van den Abeele [36] | 89 parent influencers | Belgium Netherlands | Perceptions of sharenting practices | Online survey |
| Van den Abeele, Hudders, & Vanwesenbeeck [37] | 19 kidfluencers and parents | Belgium, Netherlands, UK | Content management for children influencers | In-depth interviews |
| Van den Abeele, Vanwesenbeeck & Hudders [29] | 20 motherhood influencers | Belgium | Online risk perceptions and sharenting | In-depth interviews |
| Porfirio & Jorge [38] | 1116 posts | Portugal | Sharenting of selected Portuguese celebrities | Content analysis of social media posts |
| Esfandiari & Yao [39] | 15 parents | Iran | Sharenting | Interviews |
| Fox, Hoy, & Carter [40] | 83 fathers | USA | Father's perceptions of children privacy and willingness to share | Online survey (M-Turk); semi-structured interviews |
| Williams-Ceci et al., [41] | 246 parents | USA | Parents' perceived importance of asking children's permission before posting | Online experiment (M-Turk) |
| Walrave et al., [42] | 10 families | Belgium | Motives for parents' sharenting behavior; Adolescents' perceptions of their parents' sharenting | Semi-structured interviews |
| Jorge et al., [22] | 11 influencers | Portugal | Posting behaviour and perceptions | Content analysis of social media posts. Interviews |
| Garmendia, Martínez, & Garitaonandia [43] | 2900 children | Spain | Perceptions of sharenting | Survey |
| Holiday, Norman, & Densley [44] | 125 posts | Globally | Self-representation of parents in images of parents and children | Content analysis of posts |
| Ranzini, Newlands, & Lutz [45] | 320 parents of young children | UK | Parents' self-reported perceptions about sharing images of children | Online survey |
| Archer [23] | 45 mother influencers. 10 focus groups with non-influencer mothers | Australia | Motivations to share images of children on social media | Interviews. Focus groups |

sharenting behaviour compares to their declared privacy concerns. Consequently, this exploratory study contributes further and novel evidence to the debate on sharenting behaviour and privacy paradox in the context of motherhood influencers in the UK.

To answer the research question, we conducted a content analysis of 5,253 Instagram posts from 10 popular UK influencers, posted between August 2020 and July 2021. We then collected self-reported measures on privacy concerns and sharenting from the same sample of influencers. Below, we present the literature review and a description of the research process, followed by the data analysis and results, along with discussion and recommendations.

## Sharenting and the privacy paradox

"The 'privacy paradox' is the phenomenon where people say that they value privacy highly, yet in their behaviour relinquish their personal data for very little in exchange, or fail to use measures to protect their privacy" [31, p. 1]. This concept posits that "users have a tendency towards privacy-compromising behavior online which eventually results in a dichotomy between privacy attitudes and actual behavior" [46, p. 1039]. This topic is even more critical in the context of sharenting where images of vulnerable consumers, specifically children are being shared without their explicit consent [47]. There are assertions that influencers compromise their children's privacy rights by excessively sharing content about them with their online followers to achieve immediate gains, often conflicting with the children's right to privacy [48]. This sharing, which is integral to their brand and their connection with their audience,

creates a dynamic where they are both public figures and private individuals. On one hand, influencers derive their influence and income from the openness and authenticity they project. Followers are drawn to the perceived intimacy and transparency, believing they are getting a genuine look into the influencer's life. This level of exposure fosters trust and a sense of personal connection, which can be highly lucrative through sponsorships, partnerships, and increased engagement [49].

On the other hand, this visibility comes at a cost. The very nature of an influencer's work necessitates a level of public exposure that can infringe upon their privacy and the privacy of those who are portrayed in the influencer's content. Influencers often find themselves in situations where their personal boundaries are blurred [46]. The expectation to continuously share personal moments can lead to a sense of vulnerability and loss of control over one's private life. This paradoxical relationship between public exposure and private life creates a tension that influencers must constantly navigate and that many researchers claim to lead to experiencing a privacy paradox [29]. A considerable number of studies focused on exploring different aspects of the sharenting practice and motherhood influencers' privacy concerns. Table 1 summarises selected recent studies available in the English language academic literature on sharenting practices.

Most of those studies relied on self-reported measures of behavioural recall obtained via surveys or qualitative interviews [for example, 45,50], and primarily focused on sharenting outside of the UK. The UK was included in only two studies [37,45] compared to Belgium included in four studies. Ranzini, Newlands, and Lutz [45] used surveys with British parents to explore their perceived behaviour on Instagram and found that self-reported privacy concerns were not correlated with self-reported perceived sharenting–both concerned and unconcerned about privacy parents reported similar levels of sharenting on Instagram. Those parents who claimed they were concerned about privacy still shared as much as those who reported not being concerned, which suggested that being concerned does not decrease sharing, and provides some support for the privacy paradox. Van den Abeele, Hudders and Vanwesenbeeck [37] conducted 19 in-depth interviews with children influencers and their parents in the UK, Belgium and Netherlands to explore how the social media representations of these children influencers are managed by their parents. The authors argued that privacy paradox occurred for these parents and was followed by strategic choices with regard to how the content was created and shared on social media. Whilst useful, self-report measures (either qualitative or quantitative) of perceived behaviour are imperfect as they do not measure observed behaviour [33,51], and hence any claims about oversharing and divergence between privacy attitudes and behaviours are insufficiently supported. Where observed behaviour has been studied, the focus of the analysis has not been privacy in the context of the UK. One study employed content analysis of social media posts focused on Portugal, and another on a small sample of 125 posts globally [38,44]. Previous studies focusing on influencers asserted that privacy paradox appears but as mentioned, such studies focused on perceived rather than observed behaviour.

Our study adopts an innovative methodological approach of assessing both actual sharenting behavior and self-reported perceptions of behaviour to explore the extent to which motherhood influencers share social media content featuring children, and to explore the gap between privacy-related behaviours and opinions in the context of selected British motherhood influencers. This approach will provide a more holistic view of sharenting behaviour and privacy attitudes.

## Research method

We applied a mixed-method approach in this study: content analysis of social media posts and online survey. First, we used quantitative conceptual content analysis to collect and analyse the

content of social media posts shared by ten influencers to assess how often they share images of children on their social media pages [52]. Next, we employed an online survey to measure perceived sharing behaviour and privacy-related concerns. Whilst this methodological approach is rare, there is no theoretical or methodological reason not to apply it in this context. If we assume that there is a gap between behaviour and intentions (as argued in past studies), we propose that one way to assess it is to examine and compare both at the same time. This approach allows to explore the patterns between privacy-related observed behaviour and perceptions about sharenting practices in a novel way and addresses some of the limitations of previous studies. By integrating both methods, we gain a more comprehensive view of the phenomenon because one method may highlight gaps or inconsistencies in the other. For instance, influencers might underreport or be unaware of their actual sharenting behavior as reported in the survey, but content analysis can reveal the reality of their online activity.

We focused on Instagram and analysed posts of a sample of ten popular British motherhood influencers. In order to select a sample of influencers, we reviewed motherhood influencer 'rankings' and selected the Mother & Baby ranking to guide our selection of influencers for this study. The influencers were voted in by a poll of the top 20 influencers from 2020 and 2021 [53,54]. The motherhood influencers included in the sample had to have a personal Instagram page with a following of over 10,000 followers as this is the size of followers that enables an influencer to earn from their social media presence [55]. All influencers had to be based in the UK and their profiles were public. A total of 10 influencers was selected based on sample sizes from similar studies, as reviewed in Table 1. The influencers included in the sample were randomly chosen from the ranking of 20. The Instagram pages were checked that the content was updated at a minimum of once per week.

Instagram was selected as it is one of the most popular social media channels used by influencers and is one of the most popular social media platforms in the UK [56]. Next, a coding book was developed by both authors of the study. When deciding on the variables to measure, we followed [57] suggestion to measure four factors: 1) the amount of information shared, 2) the frequency, 3) the content and type of information shared, and 4) the audience reached by the information published about the child. In addition, we referred to a pilot study conducted in the US by [58]. The final list of coding variables in this study included: whether the post included an image of a child; the perceived age of child; whether the child's face is fully shown, partially shown, or blurred; inclusion of personal data such as name, age and location; whether the child was presented in an embarrassing situation (making fun of child, child not being clean, child posing in a silly way); whether the image appeared choreographed or natural; whether there was an intimate story shared about a child (tantrum, emotions, using a toilet, breastfeeding, illness/hospital/health-related, focus on body parts); whether the images were of an exposing nature (showing the child naked, semi-naked, or in a provocative pose); and whether the post were sponsored.

The next step was to test the coding book and descriptions of variables by coding a small sample of posts by the two coders (authors of this study). Following the testing (which involved coding together a small sample of posts), one coder conducted most of the coding. The second coder then coded the first 600 posts (11%) to ensure inter-coder reliability [59]. The coding was carried out manually by visiting each post on the influencer's profile and coding it for the presence of the specified variables. By applying this manual coding procedure, no images were therefore collected. The coding results for each post were manually entered into an SPSS file. The inter-coder reliability, measured as Cohen's kappa and calculated in SPSS for this sample, was 0.99, which is a very high inter-coder reliability [59]. Following this, the first coder coded the remainder of the sample. Next, the second coder randomly checked 5% of the coded posts. There were no disagreements in the coding of that sample apart from the perceived age of

child where there were minor discrepancies. Following this, it was decided that all images featuring children would be coded by both coders for this specific variable. The posts for a full year from August 2020 to July 2021 were analysed to take into consideration and control for seasonal fluctuations and changes.

Following the content analysis, a short questionnaire was designed to capture the influencers' perceptions about their sharenting practices. This part was carried out to compare influencers' perceived sharenting behaviour with actual sharenting behaviour as measured by content analysis of posts. We measured the following constructs: perceived sharenting behaviour [adopted from *45]; willingness to share information about child/ren [adopted from 44], and situational privacy concerns (three items) adopted from [45]. These constructs were selected based on a review of previous studies (as outlined in Table 1) as they were the most used and most reliable measures of perceived sharenting and privacy-related perceptions. S1 Table presents the constructs and items used in the questionnaire. The questionnaire was hosted on www.qualtrics.com, and the link to it was sent to the influencers included in this study via Instagram.

Ethical approval for both studies was obtained from the Liverpool Hope University Ethics Committee with the following Ethical Application IDs and dates:

Study 1: ID: 4981/ 8th July 2021 until 1st July 2022 (Covered all data collection for phase one- 8th July 2021- 28th February 2022).

Study 2: ID: 10196 14th March 2023- 3rd April 2024 (Covered all data collection for phase two- 14th March 2023- 31st July 2023).

Each study had an Ethical Application submitted which covered both data collection periods separately. For Study 1 (content analysis) the explicit consent of participants was not necessary as confirmed by the Ethics committee. For Study 2, consent was obtained via an online questionnaire by providing detailed information about the study and how data will be used and asking respondents to answer a question about their agreement to participate. Both studies involved adults only. The data collection methods for both studies complied with the terms and conditions for the source of the data as confirmed by the institutional ethical approval processes outlined above. Research datasets available from: https://openresearch.lsbu.ac.uk/item/95xy3.

## Data analysis and results—Study 1

The sample of influencers included ten British motherhood influencers. For the purposes of data analysis, we anonymised the influencers assigning them numbers from 1 to 10. Table 2 presents the characteristics of a sample. As this is an exploratory study, the small sample size is justified as the aim is not to make causal inferences and/or generalizations but to explore sharing behaviour and privacy-related opinions. The sample size is in line with past studies which relied on small numbers of influencers, for example Jorge et al. [22] examined the content of 11 influencers, and Van den Abeele, Hudders, & Vanwesenbeeck [37] examined 6 influencers per country.

Overall, 5253 posts were coded, and 3917 posts included an image or video of a child. Hence, a significant majority of the posts (74.5% of the total sample) collected from these ten motherhood influencers featured an image of a child. All studied influencers shared images of their children ranging from 43% to 99%. Whilst there are differences in the amount of shared material, all studied influencers used images of their children on their Instagram profiles.

**What do images featuring children depict?.** Majority of those images which featured a child/children, featured images of child/ren alone (56.7%, 2219 posts). Majority of posts (64%,

**Table 2. General characteristics of motherhood influencers and posts.**

| Influencer | Number of followers | Number of all posts | Does the post include a child? | | % of images featuring a child |
|---|---|---|---|---|---|
| | | | **Yes** | **No** | |
| 1 | 109,000 | 440 | 323 | 117 | 73% |
| 2 | 80,400 | 453 | 368 | 85 | 81% |
| 3 | 124,000 | 161 | 89 | 72 | 55% |
| 4 | 419,000 | 211 | 153 | 58 | 72% |
| 5 | 79,900 | 906 | 898 | 8 | 99% |
| 6 | 42,400 | 378 | 291 | 87 | 77% |
| 7 | 343,000 | 1365 | 1058 | 307 | 77% |
| 8 | 36,500 | 596 | 351 | 245 | 59% |
| 9 | 845,000 | 466 | 202 | 264 | 43% |
| 10 | 57,500 | 277 | 184 | 93 | 66% |
| **Total** | - | **5253** | **3917** | **1336** | **74.5%** |

or 2505) featured more than one child in the posts. Child/ren were shown mainly in still images with text (3536 posts or 90.3%), followed by video and text (373 posts, 9.5%). Majority of posts which featured children were perceived as natural images (3204, 81.8%). Only 8.5% of images were perceived as choreographed, but 9.8% were perceived as unclear. Almost half of the posts which featured images of children were sponsored posts–this was 1817 posts (46%). This means that in almost half of the posts, children are used as brand and product endorsers. Table 3 summarises the analysis of the general features of the posts.

**Analysis of defined privacy variables.** Table 4 provides a summary of the privacy variables and their incidence in the posts. Personal data about children was revealed in 37.6% of

**Table 3. Incidence and proportions of selected variables in the posts.**

| Variable | | Incidence (%) |
|---|---|---|
| **Does the post include a child in some way** | Yes | 3917 (74.6%) |
| | No | 1336 (25.4%) |
| | Total images | 5253 |
| **Is the child or children with their mother in the image** | Yes | 1698 (43.3%) |
| | No | 2219 (56.7%) |
| | Total | 3917 |
| **Is there more than one child in the image?** | Yes | 2505 (64%) |
| | No | 1412 (36%) |
| **What media was used to show children in the posts?** | Still image | 8 (0.2%) |
| | Still image and wording | 3536 (90.3%) |
| | Video and wording | 373 (9.5%) |
| | Total | 3917 |
| **Is it a natural or choreographed image** | Natural | 3204 (81.8%) |
| | Choreographed | 331 (8.5%) |
| | Unclear | 382 (9.8%) |
| | Total | 3917 |
| **Is it a sponsored post?** | Yes | 1817 (46%) |
| | No | 2100 (54%) |
| | Total | 3917 |

**Table 4. Incidence and proportions of selected privacy variables in the posts.**

| What personal data does the post include? | None | 2446 (62.4%) |
|---|---|---|
| | Name | 1191 (30.4%) |
| | Age | 128 (3.3%) |
| | Location | 34 (0.9%) |
| | Name and Age | 114 (2.9%) |
| | Name and Location | 4 (0.1%) |
| | Age and Location | 1 (0) |
| | Total | 3917 |
| **What parts of the child's face are shown in the image?** | Full face | 5868 |
| | Partial Face | 1589 |
| | Face hidden or blurred | 912 |
| | Total number of faces featured | 8369 |
| **Perceived age of child** | New-born | 705 |
| | Up to 1 | 980 |
| | 2 | 1004 |
| | 3 | 1325 |
| | 4 | 2222 |
| | 5 | 853 |
| | 6 | 315 |
| | 7 | 375 |
| | 8 | 236 |
| | 9 | 44 |
| | 10 | 170 |
| | 11 | 33 |
| | 12 | 68 |
| | Unknown | 39 |
| | Total | 8369 |
| **Is the child presented or discussed in an embarrassing situation?** | Embarrassing | 453 (11.6%) |
| | Not Embarrassing | 3464 (88.4%) |
| | Total | 3917 |
| **What embarrassing situation is the child involved in?** | None | 3464 (88.4%) |
| | Making fun of child | 156 (4.1%) |
| | Child not being clean | 72 (1.8%) |
| | Child posing in a silly way | 224 (5.7%) |
| | Child not clean and making fun of child | 1 (0.02%) |
| | Total | 3917 |
| **Is there an intimate or private story presented about child** | Yes | 553 (14.1%) |
| | No | 3364 (85.9%) |
| | Total | 3917 |
| **What is the content of the intimate or private story child** | None | 3364 (85.8%) |
| | Tantrums | 66 (1.7%) |
| | Emotions | 251 (6.4% |
| | Toilet | 32 (0.8%) |
| | Breast | 86 (2.2%) |
| | Illness/Hospital/Health | 97 (2.5%) |
| | Emotions and Breast | 7 (0.2%) |
| | Tantrums and Emotions | 8 (0.2%) |
| | Tantrums and Toilet | 6 (0.2%) |
| | Illness/Hospital/Health/Breast | 1 (0.02%) |
| | Body parts | 1 (0.02%) |
| | Total | 3917 |

(*Continued*)

**Table 4.** (Continued)

| What is the exposed content type | None | 3859 (98.5%) |
|---|---|---|
| | Naked | 3 (0.1%) |
| | Semi-naked | 52 (1.3%) |
| | Provocative | 3 (0.1%) |
| | Total | 3917 |

posts featuring children. In 30.4% of posts (1191) the name of the child was revealed, age was revealed in 3.3% of posts, name and age together in 2.9% of posts, and location and name and location in 1% of the posts. The posts featuring children usually featured them in non-embarrassing situations (88.4%, 3464). Those posts that featured children in embarrassing situations (453 posts), were most often shown in a 'silly pose' (224 posts, 5.7%), followed by making fun of a child (156 posts, 4.1%) and child not being clean (71 posts, 1.8%). Out of the 3917 posts featuring children, 553 (14.1%) featured an intimate story about the child, such as emotions (251 posts, 6.4%), illness-related (98 posts), breastfeeding (2.2%), and tantrums in 66 posts (1.7%). Most posts (3859, or 98.5%) did not feature any images of exposed content. Semi-naked content was featured in 52 posts, and provocative and naked in 3 posts each.

**What is the perceived age of the child/ren in the image?.** The children featured in posts were usually young children–the majority in the age group newborn up to 6-year-olds. Overall, out of 8369 instances of children presented in the 3917 posts (calculated by adding images of all children in the posts that included children), 705 (8.4%) were perceived as new-born, 980 children (11.7%) were perceived to be up to 1 year old, 1004 (12%) were aged 2, 1325 (15.8%) were aged 3, 2222 (26.5%) were aged 4. 853 (10.2%) were aged 5. 315 (3.8%) were aged 6. 965 (11.5%) images featured children older than 6 years old.

**What parts of the face are shown in the posts?.** The majority images featuring children featured the entire face of a child (over 70% of posts for each child visible in the posts). 5868 or 70% out of 8369 images of children (calculated by adding all children in the images where children were present) that were present in the images had their full faces shown in the images. 1589 or 19% of children in the images have their partial faces shown in the images. 912 (10.8%) of children in the images have their faces hidden or blurred.

**Sponsored posts.** Out of the 3917 posts that included an image of a child, almost half of the images (1817, or 46.3%) included a sponsorship (Table 5). Clothing items featured in 1048 (26.7%) images, food/drinks in 70 images, toys in 48, photography in 84, and day out/experience in 98.

**Images of children and engagement: Likes and followers.** At last, we explored the patterns between post engagement (measured by number of likes) and the number of children shown in the posts, and the number of followers and percentage of posts featuring children (Table 2). The number of followers is not positively correlated with the proportion of posts featuring children: the influencer with the highest number of followers does not have the highest proportion of images featuring children, and influencers with fewer followers do not have the lowest proportion of images with children.

Next, we calculated the correlation between likes and the number of children shown in the posts using Pearson's correlation coefficient [60]. The results showed that there is a negative correlation between the number of likes and the number of children included in the images, $r$ (3915) = -.115, $p$ = .00. This indicates that posts containing fewer children tend to garner more engagement in the form of likes. Furthermore, it implies that including children in posts is not essential for attracting followers and obtaining likes.

**Table 5. Sponsorship in posts featuring images of children.**

| Item | Number of posts | % |
|---|---:|---:|
| None | 2100 | 53.6 |
| Clothing items | 1048 | 26.7 |
| clothing and other products | 156 | 4 |
| Photography | 84 | 2.1 |
| Day out/ Experience Days | 98 | 2.5 |
| Food/Drink Products | 70 | 1.7 |
| Toys | 48 | 1.2 |
| Books | 40 | 1 |
| TV | 34 | 0.8 |
| Home Accessories | 31 | 0.8 |
| Baby Products | 39 | 1 |
| Hair | 19 | 0.5 |
| Other | 150 | 3.8 |
| **Total** | **3917** | **100%** |

## Data analysis and results- study 2

To explore the association between actual sharing behaviour and perceived sharing behaviour, and privacy attitudes, we analysed the results of the online survey and juxtaposed it with the data on sharing behaviour. Before proceeding with the data analysis, we first calculated scale reliability for Willingness to Share (Cronbach's alpha = .972) and Situational Privacy Concern (Cronbach's alpha = .919) [61]. We then calculated mean scores for the multi-item constructs of Willingness to Share and Situational Privacy Concerns. Table 6 presents the results of the survey alongside the characteristics of the posts for each of the influencers.

**Table 6. Sharenting behaviour and privacy concerns.**

| Sharenting: % of images featuring a child | Reported frequency of posting: How often did you post images of your child/ren in the past 2 years on Instagram?[1] | Willingness to share (1–5) | Situational Privacy Concern score (1–5)[2] |
|---:|---|---|---|
| 99% | 2–3 days per week* | 2.79 (Not willing/ Indifferent) | 4.33 (Not concerned) |
| 81% | 4–6 days per week** | 3.14 (Indifferent/willing) | 4 (Not concerned) |
| 77% | 2–3 days per week* | 3.71(Indifferent/willing) | 4.33(Not concerned) |
| 77% | 2–3 days per week* | 3.07 (Indifferent/willing) | 3 (Indifferent) |
| 73% | 4–6 days per week** | 3.21(Indifferent/willing) | 4 (Not concerned) |
| 72% | 4–6 days per week** | 3.43 (Indifferent/willing) | 5 (Not concerned at all) |
| 66% | 4–6 days per week** | 3.43 (Indifferent/willing) | 3.67 (Indifferent)) |
| 59% | Once per week* | 3.07 (Indifferent/willing) | 4.33 (Not concerned) |
| 55% | 2–3 days per week* | 3.5 (Indifferent/willing) | 4 (Not concerned) |
| 43% | NA | NA | NA |

[1] Once per week–amounts to 14% of posts featuring children; 2–3 days per week–amounts to 28–44% posts featuring children, 4–6 days per week–amounts to 57–85% posts featuring children; every day–amounts to 100% posts featuring children.

*Not accurately estimated posting behaviour.

** Accurately estimated posting behaviour.

[2]For better readability of the results for this question we translated the scale points in this table in the following way: Strongly disagree–Very concerned; Disagree–Concerned, Neither agree Nor Disagree–Indifferent, Agree–Not concerned, Strongly Agree–Not Concerned at all).

When asked to report the frequency of posting images of their children online, only four (out of nine) influencers accurately estimated their sharenting behaviour, and five did not accurately estimate their sharenting behaviour (Table 6). In order to calculate this, we compared the proportion of images featuring children shared by each influencer (Sharenting) to the reported frequency of posting for that influencer. The reported frequency of posting (once per week, 2–3 days per week, 4–5 days per week, every day) was transformed into percentages based on the rule that 7 days per week is 100%, and 1 day per week amounts to 14%. For example, if an influencer featured children in 99% of the posts this was assumed to mean that this influencer posted a post featuring an image almost every day. This was then compared to the reported frequency of posting for each influencer. If the reported frequency closely matched the actual sharing behavior, it was considered an accurate estimate of posting behavior. If the reported frequency did not closely match the actual sharing behavior, it was rated as an inaccurate estimate. Our analysis suggests that self-reported perceived behaviour may not be the most accurate measure of assessing actual sharing behaviour and may not be the most suitable methodological approach to examine behaviour.

When asked about willingness to share their children's images online, 8 out of 9 of the influencers reported being either indifferent or willing to share images of their children online. Moreover, all nine influencers reported feeling safe online (Situational Privacy Concern score) and did not perceive sharenting as a threat to theirs and their children's privacy. This suggests that sharing images of children by these influencers may be strategic rather than accidental. These influencers seem to choose to share images and details about their children consciously. This may be specific to this sample, as motherhood influencers promote motherhood, and children are an integral part of that 'product'.

## Discussion

The primary aim of this study was to investigate the extent to which selected British motherhood influencers disclose private information about their children on Instagram and how this behaviour correlates with their self-reported perceptions of such sharing. We analysed over 5000 Instagram posts to measure the sharenting behaviour of these influencers and conducted an online questionnaire to capture their views on online privacy and their perceived sharenting behaviour.

The results show that all motherhood influencers share images of their children, though the frequency varies. A significant majority of posts (74.6%) featured an image of a child, and 68.9% of these displayed the child's full face. The presence of children in posts ranged from 43% to 99% across individual influencers. Most child-featured posts were natural images indicating a priority on authentic portrayals of motherhood experiences [62]. Children's real names were revealed in 20% of posts, raising concerns about the implications for protecting a child's identity and the potential negative future consequences [63]. Additionally, 9.85% of posts depicted children in embarrassing situations, with the most common themes being "making fun of the child" and the "child presented in a silly pose." Intimate images and stories accounted for 11.2% of posts, while exposing images comprised 7.38%. Despite these relatively low proportions, the ethical question remains whether it is ever acceptable to share images of children who cannot consent to their online presence.

Interestingly, there was a negative correlation between posts featuring more children and the number of likes received. This finding has important positive implication for children's privacy: while it may not be feasible to completely exclude children from posts given the family focus of their content, it is still possible to maintain sponsorship opportunities and a strong following without always including images of children. Nearly half of all posts were sponsored or

linked to a product or service, indicating that sponsorship opportunities often outweigh privacy concerns [23,25]. Consequently, children's images are used for financial gain, likely without their informed consent, as children cannot fully understand the potential consequences of such actions.

Relating the content analysis to survey results, we discuss these findings in the context of the privacy paradox. Most motherhood influencers scored relatively high on the Situational Privacy Concern measure, with seven out of nine feeling confident that their children were safe despite the online content shared about them. These results indicate that motherhood influencers do not perceive online sharing of their children's images as a threat to their safety. Most influencers were either indifferent or willing to share such content. Only one influencer's responses diverged from their actual sharing behavior.

This consistency between actual sharenting behavior and declared privacy views is in contradiction with past research that reported internal conflicts among parent influencers regarding how much to share online. For example, Blum-Ross & Livingstone [64] suggest that motherhood influencers struggle between keeping their children's lives private and presenting them in posts due to follower expectations for relatable content. However, our study suggests that the influencers in our sample are very aware of their sharenting behavior and do not perceive sharing images of their children as a future threat. Similar observations were made by Van den Abeele, Hudders, & Vanwesenbeeck [65], who found that mothers of child influencers made conscious decisions about the frequency and type of content to post.

## Theoretical discussion

Theoretically, this study contributes to a deeper understanding of the privacy paradox in the context of British motherhood influencers. By employing a mixed-methods approach to combine two measures of behaviour (observed and self-reported), we delineate the relationship between privacy concerns and online sharing behaviour among selected British motherhood influencers. Our findings suggest that the privacy paradox is not strongly supported by this study's results, lending support to the argument that the privacy paradox is not always present and is context-specific, rather than universally applicable to all online environments [31,66]. The influencers in our study appear to be aware of their online sharing behaviour and are confident that their sharenting will not result in any threats stemming from their presence on social media platforms. Our findings suggest that children's privacy may be intentionally compromised due to the commercial nature of these influencers, and their willingness to relinquish privacy for gratification and financial gain is deliberate [67]. The study does not provide strong support for the privacy paradox, indicating that this phenomenon may vary by individual [35], be context-dependent or research-method dependent [31] or subject to change over time [35] as consumers adapt to new technologies and leverage them [16], or where concerns about privacy may not be significant in contexts where gratification is readily obtained [68]. In addition, the privacy paradox is said to apply to individuals, but social media influencers who monetize their social media content are commercial entities rather than individuals. For example, Ranzini, Newlands & Lutz [45] found that privacy concerns were unrelated to sharenting behaviour–perhaps because of the commercial nature of sharenting and the fact that such sharenting is performed strategically with parents deliberately using children's images and details to benefit financially. In another interview-based study [37], mother influencers shared that they believed that portraying children in social media content is essential to enhance credibility, authenticity, and intimacy, but they also believed that such perceptions can be achieved while protecting the child's privacy by following privacy protecting techniques. Another explanation may be that motherhood influencers, as suggested by Van den Abeele, Vanwesenbeeck, and

Hudders [65] do not perceive their sharenting behaviour as privacy risk because such risk is relatively abstract and distant because the majority of them have not (yet) personally experienced any negative consequences of sharing information about their children.

## Practical and policy implications

The results in this study demonstrate that motherhood influencers use images of their children in majority of social media posts, and in almost half sponsored posts suggesting that children's images are used for financial gain without their consent. When the content analysis results are integrated with the self-reported data, it appears that the influencers in this study deliberately share images of their children and feel confident in doing so. This suggests that sharing may occur strategically (by choice) rather than accidentally. Although sensitive content was shared relatively sporadically, we join the call to advocate that new legislation should be developed at governmental level to protect children online and safeguard them to prevent them from being taken advantage of [69]. Whilst greater limitations on user-generated content on social media are being introduced, the rights of children are not yet sufficiently specified in the legislation. "Young digital labor" [70,71] such as featuring children on professional influencers' profiles and sponsored posts as evidenced in this study seems to be strategically practiced by the motherhood influencers. Pay and conditions for children are not presently taken into consideration for children presented in influencers posts and thus rules are minimal to protect child privacy in the UK, potentially leaving children in this field vulnerable to exploitation [69]. Dobson & Jay [63] propose that children should be viewed on social media as being able to hold their own rights to privacy, manage their own self-expression and overall be able to have a voice. Therefore, it is thought that the perspective of children should always be considered, especially as social media posts may influence the development of their own self-identities and self-image which can potentially conflict with what parents choose to disclose and share on social media [44,72]; and that labour of children should be restricted to a specific number of hours and renumerated [37]. However, we argue that, as evidenced in our study, sharenting is performed by mothers as a strategic activity, with mothers willing to share images of their children publicly. This suggests that mothers may not be aware of the potential future negative consequences of such sharing. We argue that if children are seen as not being able to recognize harmful advertising for crisps, and therefore their exposure to such advertisements is restricted [73,74], they are even less able to recognize the social and psychological long-term consequences of their online activities, or their content being shared online. Hence, suggestions to consider their 'voices' in sharenting are misguided especially because, as noted before, children "derive little to no benefits from their mothers' influencer activities, yet are the ones carrying the potential privacy risks" [29, p. 297]. Our study suggests that sharenting may be strategic and therefore suggests that mothers may not even experience any privacy-related discomfort which provides even more support for more content regulation. We therefore suggest policies strictly regulating the content shared publicly online [75]. For example, social media companies should take more responsibility for the safeguarding of children and implement their own platform-based rules [76]. This can go as far as introducing legislation that, as Solove [31] suggested changes "the architecture that structures the way information is used, maintained, and transferred" (p. 6), such as for example a ban on featuring children in influencers' posts or developing or modifying social media apps to automatically detect and block images of children.

In managerial terms, this research indicates that there is no positive correlation between popularity (measured by follower count) and engagement (likes) when there is an increase in the number of posts featuring children. This suggests that popularity can be achieved without

using children as co-influencers, and as evidenced by a study by Chung, Ding & Kalra [49] who found that engagement was positively correlated with number of people in general. Additionally, despite the low percentages of embarrassing, intimate, and exposing content found in this study, the influencers sampled still maintain a high number of followers. This suggests that there might be less pressure on influencers to present their children in compromising ways to gain acceptance from supporters, which is often perceived as a necessary strategy for motherhood influencers.

## Future research and limitations

Whilst this study addressed an important gap, it should be viewed within the following limitations. First, the study is exploratory and descriptive in nature and based on a small sample of British motherhood influencers, and therefore any insights generated by this study should be further verified in further exploratory and confirmatory studies based on larger samples. The sample of posts does not include content shared via 'stories' option as statistics on stories are not publicly available, but stories could reveal more about sharing images of children because stories appear on Instagram for a limited period. However, including stories in any study would require cooperation from the influencers to share data about stories engagement. Our study focused on a relatively small sample of larger influencers, but it would also be important to examine whether nano-influencers (under 10,000 followers) share more details about their children due to their desire to gain followers. Another important limitation relates to the design of the questionnaire. Specifically, in the introduction, participants were informed that the study focused on sharenting behaviour and privacy perceptions, which may have influenced their responses. Future research could explore the impact of such disclosure on reported privacy perceptions. In addition, at the time of data collection, 'likes' served as the standard engagement measure. However, since the study was conducted, influencers and other posters on social media have the ability to hide likes. This is to improve social media users' well-being and depressurise social media in general [77]. This should be considered in future studies, as alternative forms of engagement may need to be explored, or the inclusion criteria may need to specify the types of engagement metrics used.

Moreover, he data was collected during the COVID– 19 pandemic and this needs to be considered when interpreting the results, as it may have had an impact on the type of content and frequency of sharing (for example increased/decreased frequency of sharing due to the introduction of remote working for many working parents).

The topic could also be explored from a gender and cross-cultural perspectives. Specifically, a future study could consider the incidence of sharing behaviour for fatherhood influencers and influencers representing different cultures and ethnicities. Finally, other social media platforms, such as TikTok or Weibo could also be studied.

## Supporting information

**S1 Table. Questionnaire items.**
(DOCX)

## Author Contributions

**Conceptualization:** Katherine Baxter, Barbara Czarnecka.

**Formal analysis:** Katherine Baxter, Barbara Czarnecka.

**Investigation:** Katherine Baxter, Barbara Czarnecka.

**Methodology:** Katherine Baxter, Barbara Czarnecka.

**Resources:** Katherine Baxter.

**Writing – original draft:** Katherine Baxter, Barbara Czarnecka.

**Writing – review & editing:** Katherine Baxter, Barbara Czarnecka.

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
