## [Decision Letter · Decision Letter 0]

1 Oct 2024

PONE-D-24-24245British motherhood influencers and sharing images of children on Instagram – do they share more than they intend to?PLOS ONE

Dear Dr. Baxter,

Thank you for submitting your manuscript to PLOS ONE. After careful consideration, we feel that it has merit but does not fully meet PLOS ONE’s publication criteria as it currently stands. Therefore, we invite you to submit a revised version of the manuscript that addresses the points raised during the review process.

We look forward to receiving your revised manuscript.

Kind regards,

Vincenzo Auriemma

Academic Editor

PLOS ONE

**Journal Requirements:**

2. In your Methods section, please include additional information about your dataset and ensure that you have included a statement specifying whether the collection and analysis method complied with the terms and conditions for the source of the data.

**Additional Editor Comments:**

Dear author,

given the comments received it is recommended to follow them and make the requested changes.

Reviewers' comments:

Reviewer's Responses to Questions

**Comments to the Author**

1. Is the manuscript technically sound, and do the data support the conclusions?

Reviewer #1: Yes

Reviewer #2: Partly

2. Has the statistical analysis been performed appropriately and rigorously? 

Reviewer #1: Yes

Reviewer #2: I Don't Know

3. Have the authors made all data underlying the findings in their manuscript fully available?

Reviewer #1: Yes

Reviewer #2: Yes

4. Is the manuscript presented in an intelligible fashion and written in standard English?

Reviewer #1: Yes

Reviewer #2: Yes

5. Review Comments to the Author

**Reviewer #1:** The introduction of the manuscript is clear, but the transition to the methodology could benefit from more explanation. Specifically, it would help to clarify why the mixed-method approach, combining content analysis and a survey, was chosen. Briefly discussing the research gaps that led to this choice would strengthen the methodology’s rationale.

The research methods section is detailed but would be improved by justifying the sample size of ten influencers. Explaining why this sample is sufficient for an exploratory study, perhaps by referencing similar research, would enhance its credibility. Additionally, the coding process is well-described, but it would be useful to specify if the coding variables were derived from prior research or developed specifically for this study. This would provide stronger links to existing literature and show how the study addresses key privacy concerns.

The criteria for selecting influencers, such as follower count and rankings, should also be explained further. A brief statement on why these criteria were relevant—such as their commercial impact or influence, would add depth. Similarly, the time frame for the content analysis could be better justified. Was it chosen to capture specific trends or seasonal variations, or due to the effects of the COVID-19 pandemic? This would give more context to the data.

The manuscript discusses inter-coder reliability, but it would be helpful to include a brief note on how disagreements between coders were resolved. This would reinforce the reliability of the analysis. The survey component would also benefit from a clearer explanation of the reasoning behind the selected questions and how they complement the content analysis, adding transparency to the method.

In the results section, the link between the analysis and the chosen methods could be clearer. Explaining why specific statistical tools were used, such as Pearson correlation, would confirm that the methods were appropriate for this type of research. Additionally, the relationship between the survey and content analysis should be clarified earlier, explaining how the two methods complement each other for a fuller understanding of sharenting behavior.

**Reviewer #2:** Dear Authors,

Thank you for the opportunity to review your manuscript. Overall, I found the topic and approach highly relevant and innovative. The coding of such a large sample is particularly impressive, and I commend you for the effort and meticulousness this must have required. However, after thoroughly reading your work, I have several questions and suggestions that I believe could further improve the clarity and rigor of the paper.

First, I noticed that in your reference table, you cite the study by Beuckels & De Wolf as an online survey, whereas it is actually a literature review. Perhaps you meant to reference Beuckels, E., Hudders, L., Vanwesenbeeck, I., & Van den Abeele, E. (2024), which focuses on the role of children and privacy management within influencer content.

The problem statement is well-articulated, and your methodological approach is clearly innovative. However, I found the articulation of your research questions lacking. A clear set of research questions or hypotheses would greatly benefit the reader in terms of understanding the scope of the study. Following a different methodological approach, as mentioned, is not a research question in itself. Given the existing body of research on this topic and your quantitative approach, I believe it would be more appropriate to frame the study around clear hypotheses. Doing so would clarify your expectations regarding the variables of interest and ensure a stronger alignment between the literature review, methodology, and results. At present, several variables and analyses emerge in the methodological section without having been introduced or discussed in the literature review, which can make it difficult for the reader to follow the rationale behind their inclusion. By adopting a hypothesis-driven approach and ensuring that all key variables are grounded in the literature, you can create a more cohesive narrative. This shift could potentially improve the flow of the manuscript but also help emphasize the theoretical contributions of your work, which remain somewhat unclear in the current draft beyond the introduction of a new methodological approach.

Additionally, I am unclear about how you arrived at your sample of ten influencers for the content analysis. Were there only ten out of twenty who met your inclusion criteria, or did you have to make additional choices? Clarifying this process would be helpful.

The section on scraping and ethics also needs more detail. You mention that active consent was unnecessary for Study 1, but this issue is under ongoing debate within the field. Why did you determine consent was unnecessary, and how did you manage the data? For example, were images saved or simply coded directly from the platform? A deeper discussion of these ethical considerations is necessary.

Additionally, in Study 2, did the mothers know that their profiles had been previously analyzed? Could this awareness have influenced their responses, possibly explaining why the privacy paradox did not emerge? Is it possible that knowing their profiles had been examined made them more conscious of their sharenting behavior, whereas they might not have been without this priming?

Regarding your inter-coder reliability, how was this tested in SPSS? A more detailed report of the analysis and results would be beneficial.

The organization of the methodological section could also be clarified. It is not immediately clear that the questionnaire is a second study, as it follows the content analysis without clear separation. Consider using distinct subheadings or treating it as one multi-method study to enhance the manuscript’s readability.

Your data collection dates back to 2021, when influencers were still displaying like counts publicly on Instagram. However, with many influencers now opting to hide likes, it may be worth discussing the relevance of using 'likes' as an engagement metric today. While the 2021 data remains valid and provides valuable insights into which content generated higher engagement, reflecting on current platform practices in your discussion would enhance the contemporary relevance of your findings.

In your analysis, I’m curious about your choice to focus on the number of children and its correlation with likes. It seems that the preference for depicting intimate social connections, as highlighted in Chung, J., Ding, Y., & Kalra (2023), could explain this correlation as well. Perhaps it’s not the number of children but rather the number of people in general that is driving engagement? In that sense, just comparing the impact of the presence vs. absence of children in posts could be more insightful, but overall, I wonder whether this would yield meaningful insights into sharenting behaviors as your overall sample does oftentimes engage in sharenting. After all, an influencer who typically features their children may also post content without them for reasons unrelated to privacy or anti-sharenting, which may not prompt strong follower reactions. Therefore, can you tell much about the impact of sharenting (vs. not sharenting) with this current sample of influencers?

I am also confused by the reversal of your privacy concern scale, as mentioned in your footnote. A high score would typically indicate a high concern, so more explanation on this reversal would be helpful.

Furthermore, your discussion on influencers underestimating or correctly estimating their sharenting frequency requires further clarification. How did you analyze this, especially given the differences in how you report the 'percentage of sharenting' versus their 'self-estimated frequency in days' in Table 6? Clearer reporting of your statistical analyses, such as whether you used sum scales or single items, would improve the rigor of your results section. Additionally, the analyses themselves (e.g., correlations, Cronbach’s alpha for scales) are not reported, which is typically expected in quantitative studies and would strengthen the transparency and reliability of your findings.

Some of your results appear to be somewhat overstated based on what was actually measured. For instance, you state: “Moreover, all nine influencers reported feeling safe online (Situational Privacy Concern score) and did not perceive sharenting as a threat to theirs and their children’s privacy.” However, it seems the single item you reference here only asks whether they perceive their “presence on Instagram” as a threat to their children’s privacy, rather than their specific sharenting behavior. Furthermore, you conclude that their sharenting is most likely strategic rather than accidental based on this, but I feel there is a gap in the reasoning. Additionally, I question whether the three items you use (only one of which directly relates to sharenting) are sufficient to fully refute the privacy paradox?

The discussion section would also benefit from more direct engagement with your results. For instance, what does it mean that the number of followers and the extent of sharenting are not correlated? Are these findings consistent with the literature? Without explicit research questions in your literature review, it is difficult to gauge the significance of your results in relation to existing studies.

Lastly, I question the focus exclusively on mothers. There is a growing body of research on ‘dadfluencers,’ ‘Instadads,’ and parent influencers more broadly. Is it necessary to limit the study to mothers, or does this reinforce gendered power dynamics? The current focus seems to place disproportionate blame on mothers, potentially increasing pressure on them in a culture where individualistic parenting and intensive motherhood is already the norm and increasingly results into parental burnouts worldwide (cf. Roskam, I., Aguiar, J., Akgun, E., Arikan, G., Artavia, M., Avalosse, H., ... & Mikolajczak, M. (2021). Parental burnout around the globe: A 42-country study. Affective science, 2(1), 58-79.)

In summary, I believe your study makes a valuable contribution to the field, but it would benefit from greater detail, clearer alignment between your research aims and results, and a more nuanced discussion.

6. PLOS authors have the option to publish the peer review history of their article (what does this mean?). If published, this will include your full peer review and any attached files.

Reviewer #1: No

Reviewer #2: No

---

## [Author Response · Author response to Decision Letter 0]

5 Nov 2024

Dear Editor,

Dear Reviewers,

Thank you for giving us the opportunity to revise and resubmit our manuscript.

Special thanks to the two reviewers who provided detailed comments on our manuscript. We acknowledge the time and effort that has been invested in reviewing our manuscript. 

We have responded to the comments (as we understood them) in the table below. Please use the ‘PLOS ONE Anon Manuscript with tracked changes’ to review the changes. 

Thank you,

Authors

Reviewer 1

Comments

The introduction of the manuscript is clear, but the transition to the methodology could benefit from more explanation. Specifically, it would help to clarify why the mixed-method approach, combining content analysis and a survey, was chosen. Briefly discussing the research gaps that led to this choice would strengthen the methodology’s rationale.

Thank you for your comment. We re-read the Introduction section and added additional explanation to further underline the choice of content analysis and survey as a mixed -method approach. Content analysis examined behaviours, and survey measured perceptions, together they address the methodological gap in previous studies. Please see Lines 73- 90 for further clarification. We also added further arguments in Lines 167- 171 how these two methods complement each other. 

The research methods section is detailed but would be improved by justifying the sample size of ten influencers. Explaining why this sample is sufficient for an exploratory study, perhaps by referencing similar research, would enhance its credibility. Additionally, the coding process is well-described, but it would be useful to specify if the coding variables were derived from prior research or developed specifically for this study. This would provide stronger links to existing literature and show how the study addresses key privacy concerns.

Thank you for your comment. The coding variables were mainly extracted from McTigue (2021)[58] who conducted a pilot study on the subject. This process is described in lines 185 - 197. The sources that the coding framework is based on were referenced in the original submission. In addition, we added a sentence (lines 246-249) to show past studies which also relied on a small number of influencers. 

The criteria for selecting influencers, such as follower count and rankings, should also be explained further. A brief statement on why these criteria were relevant—such as their commercial impact or influence, would add depth. Similarly, the time frame for the content analysis could be better justified. Was it chosen to capture specific trends or seasonal variations, or due to the effects of the COVID-19 pandemic? This would give more context to the data.

We have explained the reason why certain influencers were selected in Lines 179-181 and later 210-212 (we analysed a year’s worth of social media images as this allows for controlling seasonal fluctuations). 

We mentioned the fact that the data collected covered some of the Covid 19 pandemic to make readers aware of the context of the data collection - however, in the revised manuscript we moved this sentence to Limitations. 

The manuscript discusses inter-coder reliability, but it would be helpful to include a brief note on how disagreements between coders were resolved. This would reinforce the reliability of the analysis. The survey component would also benefit from a clearer explanation of the reasoning behind the selected questions and how they complement the content analysis, adding transparency to the method.

We added further details about the coding process and how we dealt with minor disagreements in Line 198-210. 

Lines 220- 222 explain why the variables for the questionnaire were chosen. 

In the results section, the link between the analysis and the chosen methods could be clearer. Explaining why specific statistical tools were used, such as Pearson correlation, would confirm that the methods were appropriate for this type of research. Additionally, the relationship between the survey and content analysis should be clarified earlier, explaining how the two methods complement each other.

We clarified that we used Pearson’s correlation coefficient to calculate the correlation between likes and number of children featured in the posts (Line 315).

We also clarified that we calculated scale reliabilities using Cronbach’s alphas (Line 323-327), added a clarification of how we coded the estimation of posting behaviour and perceived posting behaviour (Lines 330 - 3451. 

Lines 167-171 explain how the two methods provide different types of data related to sharenting behaviour and sharenting behaviour perceptions. 

Reviewer 2

First, I noticed that in your reference table, you cite the study by Beuckels & De Wolf as an online survey, whereas it is actually a literature review. Perhaps you meant to reference Beuckels, E., Hudders, L., Vanwesenbeeck, I., & Van den Abeele, E. (2024), which focuses on the role of children and privacy management within influencer content.

Apologies this was a mistake and indeed we meant to cite Beuckels, E., Hudders, L., Vanwesenbeeck, I., & Van den Abeele, E. (2024).

This has now been updated in the table. 

The problem statement is well-articulated, and your methodological approach is clearly innovative. However, I found the articulation of your research questions lacking. A clear set of research questions or hypotheses would greatly benefit the reader in terms of understanding the scope of the study. Following a different methodological approach, as mentioned, is not a research question in itself. Given the existing body of research on this topic and your quantitative approach, I believe it would be more appropriate to frame the study around clear hypotheses. Doing so would clarify your expectations regarding the variables of interest and ensure a stronger alignment between the literature review, methodology, and results. At present, several variables and analyses emerge in the methodological section without having been introduced or discussed in the literature review, which can make it difficult for the reader to follow the rationale behind their inclusion. By adopting a hypothesis-driven approach and ensuring that all key variables are grounded in the literature, you can create a more cohesive narrative. This shift could potentially improve the flow of the manuscript but also help emphasize the theoretical contributions of your work, which remain somewhat unclear in the current draft beyond the introduction of a new methodological approach.

Thank you for your comment. As this was an exploratory study we did not state any hypotheses. 

If we were to design a confirmatory study with hypotheses, the research design process and everything that follows would probably be very different. Our position is that we would like to keep it as an exploratory study rather than to re-write it as a confirmatory study. 

However, we did update the phrasing of the Introduction to better articulate the aims of the study in Lines 77-90. 

However, we do agree that a confirmatory study would be beneficial in the future and mentioned the exploratory and descriptive nature of this study as a limitation in the ‘Future research and limitations’ section. 

Additionally, I am unclear about how you arrived at your sample of ten influencers for the content analysis. Were there only ten out of twenty who met your inclusion criteria, or did you have to make additional choices? Clarifying this process would be helpful.

We elaborated on the selection of 10 influencers in lines 178-182. 

We added a sentence (lines 247-249) to show past studies which also relied on a small number of influencers as these past studies guided our sample size selection. 

The section on scraping and ethics also needs more detail. You mention that active consent was unnecessary for Study 1, but this issue is under ongoing debate within the field. Why did you determine consent was unnecessary, and how did you manage the data? For example, were images saved or simply coded directly from the platform? A deeper discussion of these ethical considerations is necessary.

We did not scrape any content from Instagram, therefore the images were not saved, they were directly coded from each profile of the social media profiles. We described this process in lines 198-204. We also added in Line 179 the following phrase to clarify that we chose only public profiles: “and their profiles were public”. 

Because we did not collect any social media images, our Ethics committee did not raise any issues related to consent. 

However, we conducted the study anonymising the participants and presenting no identifiable information. 

Additionally, in Study 2, did the mothers know that their profiles had been previously analyzed? Could this awareness have influenced their responses, possibly explaining why the privacy paradox did not emerge? Is it possible that knowing their profiles had been examined made them more conscious of their sharenting behavior, whereas they might not have been without this priming?

Thank you for this comment. We think it would be very interesting to study the role of priming in the privacy paradox. Indeed, priming may influence how we report our behaviour. 

At present, we do not know whether priming would exacerbate either responses or dampen it so this is an interesting area for future research. 

The participants were informed that we were exploring sharenting behaviour and their attitudes towards privacy. We added a sentence to describe this in the Limitations section (Lines 502 - 512).

Regarding your inter-coder reliability, how was this tested in SPSS? A more detailed report of the analysis and results would be beneficial.

In Lines 204-206, we added a clarification that Cohen’s kappa was calculated in SPSS ( we did not describe the procedure step by step as this is not usually the procedure for reporting analysis, but we followed this process: https://statistics.laerd.com/spss-tutorials/cohens-kappa-in-spss-statistics.php (our variables were nominal). 

The organization of the methodological section could also be clarified. It is not immediately clear that the questionnaire is a second study, as it follows the content analysis without clear separation. Consider using distinct subheadings or treating it as one multi-method study to enhance the manuscript’s readability.

We have added a separate heading separating study two. 

Your data collection dates back to 2021, when influencers were still displaying like counts publicly on Instagram. However, with many influencers now opting to hide likes, it may be worth discussing the relevance of using 'likes' as an engagement metric today. While the 2021 data remains valid and provides valuable insights into which content generated higher engagement, reflecting on current platform practices in your discussion would enhance the contemporary relevance of your findings.

We added an explanation regarding likes in Lines 502 - 507. 

In your analysis, I’m curious about your choice to focus on the number of children and its correlation with likes. It seems that the preference for depicting intimate social connections, as highlighted in Chung, J., Ding, Y., & Kalra (2023), could explain this correlation as well. Perhaps it’s not the number of children but rather the number of people in general that is driving engagement? In that sense, just comparing the impact of the presence vs. absence of children in posts could be more insightful, but overall, I wonder whether this would yield meaningful insights into sharenting behaviors as your overall sample does oftentimes engage in sharenting. After all, an influencer who typically features their children may also post content without them for reasons unrelated to privacy or anti-sharenting, which may not prompt strong follower reactions. Therefore, can you tell much about the impact of sharenting (vs. not sharenting) with this current sample of influencers? 

We agree that the presence of people rather than just children may be an important factor. However the focus of the paper was to explore the correlations between images of children and engagement (measured by the number of likes) - and our results show that the number of children in posts does not correlate with likes. 

We agree that another study to compare engagement effects of sharenting versus no no-sharenting would be useful, perhaps an experimental study. 

I am also confused by the reversal of your privacy concern scale, as mentioned in your footnote. A high score would typically indicate a high concern, so more explanation on this reversal would be helpful.

We did not reverse the scale. S 1 (Table) shows the original scale we used (1 - Strongly disagree and strongly agree - 5). We labelled the results in Table 6 so that the reader does not need to go back to table S1 to decode the scores. High score indicates ‘no concern’. Here are the questions:

1. Overall, I see no real threat to the privacy of my children due to my presence on Instagram 

2. I know that nothing unpleasant will happen to my children due to my presence on Instagram

3. Overall, I find it safe to publish my child/ren’s personal information on Instagram

Respondents had to choose an option of Strongly disagree 1 - strongly agree 5. So the higher the score the smaller the concern - hence our labelling of scores of 4 as ‘no concern’ (in Table 6). 

Furthermore, your discussion on influencers underestimating or correctly estimating their sharenting frequency requires further clarification. How did you analyze this, especially given the differences in how you report the 'percentage of sharenting' versus their 'self-estimated frequency in days' in Table 6? Clearer reporting of your statistical analyses, such as whether you used sum scales or single items, would improve the rigor of your results section. Additionally, the analyses themselves (e.g., correlations, Cronbach’s alpha for scales) are not reported, which is typically expected in quantitative studies and would strengthen the transparency and reliability of your findings.

We added note 1 to Table 6 to explain how we coded the Reported frequency of posting and how this was compared to actual sharenting behaviour.

In Lines 330 - 351, we further explained how we compared the Sharenting behaviour to Reported frequency of posting. 

We added Cronbach’s alphas for the multi-item scales (Lines 323- 327). 

We did not calculate correlations between variables - because this is typically done for regression analysis and not for descriptive exploratory studies. We understood the reviewer’s request related to correlations between variables as one relating to checking multicollinearity for regression analysis. 

Some of your results appear to be somewhat overstated based on what was actually measured. For instance, you state: “Moreover, all nine influencers reported feeling safe online (Situational Privacy Concern score) and did not perceive sharenting as a threat to theirs and their children’s privacy.” However, it seems the single item you reference here only asks whether they perceive their “presence on Instagram” as a threat to their children’s privacy, rather than their specific sharenting behavior. Furthermore, you conclude that their sharenting is most likely strategic rather than accidental based on this, but I feel there is a gap in the reasoning. Additionally, I question whether the three items you use (only one of which directly relates to sharenting) are sufficient to fully refute the privacy paradox?

Our intention was not to refute the privacy paradox but rather provide a new perspective by applying a different methodological approach. We read again the Discussion part and made sure we do not use any overstatement and only state that the privacy paradox is not supported by the results of this particular study - we cannot possibly refute the privacy paradox with just one study of this size and it was not our intention. We merely wanted to verify and offer further suggestions on how to research the privacy paradox. 

Our claim that it is strategic rather than acci

---

## [Editor Report · Decision Letter 1]

12 Nov 2024

Sharing images of children on social media: British motherhood influencers and the privacy paradox.

PONE-D-24-24245R1

Dear Dr. Baxter,

We’re pleased to inform you that your manuscript has been judged scientifically suitable for publication and will be formally accepted for publication once it meets all outstanding technical requirements.

Kind regards,

Vincenzo Auriemma

Academic Editor

PLOS ONE
---

## [Editor Report · Acceptance letter]

27 Nov 2024

PONE-D-24-24245R1 

PLOS ONE

Dear Dr. Baxter, 

I'm pleased to inform you that your manuscript has been deemed suitable for publication in PLOS ONE. Congratulations! Your manuscript is now being handed over to our production team.

Kind regards, 

on behalf of

Dr. Vincenzo Auriemma 

Academic Editor

PLOS ONE